# Molecular basis of MHC I quality control in the peptide loading complex

Alexander Domnick [1,3], Christian Winter [1,3], Lukas Sušac[1], Leon Hennecke[1], Mario Hensen [2], Nicole Zitzmann [2], Simon Trowitzsch [1], Christoph Thomas [1] & Robert Tampé [1] ✉

Major histocompatibility complex class I (MHC I) molecules are central to adaptive immunity. Their assembly, epitope selection, and antigen presentation are controlled by the MHC I glycan through a sophisticated network of chaperones and modifying enzymes. However, the mechanistic integration of the corresponding processes remains poorly understood. Here, we determine the multi-chaperone-client interaction network of the peptide loading complex (PLC) and report the PLC editing module structure by cryogenic electron microscopy at 3.7 Å resolution. Combined with epitope-proofreading studies of the PLC in near-native lipid environment, these data show that peptide-receptive MHC I molecules are stabilized by multivalent chaperone interactions including the calreticulin-engulfed mono-glucosylated MHC I glycan, which only becomes accessible for processing by α-glucosidase II upon loading of optimal epitopes. Our work reveals allosteric coupling between peptide-MHC I assembly and glycan processing. This inter-process communication defines the onset of an adaptive immune response and provides a prototypical example of the tightly coordinated events in endoplasmic reticulum quality control.

The adaptive immune system crucially depends on cell surface presentation of the host proteome in the form of peptides on MHC I molecules[1–3]. These peptide-MHC I (pMHC I) complexes are sampled by cytotoxic T cells, which elicit apoptosis of target cells when non-self peptides are identified[4]. Kinetically stable pMHC I complexes presenting immunogenic peptides are necessary to mount an efficient and specific immune response[5,6]. Kinetically stable pMHC I complexes are ensured by stringent peptide proofreading and quality control processes[7–9], which are orchestrated by the peptide loading complex (PLC), a highly dynamic multi-chaperone machinery in the endoplasmic reticulum (ER)[10]. As central part of the PLC, the transporter associated with antigen processing (TAP) ensures a high local concentration of translocated peptides in the ER lumen proximal to the PLC[1,5,10,11]. The other PLC constituents, namely calreticulin, tapasin, and ERp57, function in recruiting and stabilizing peptide-receptive MHC I

clients, whereby tapasin serves an additional outstanding role as peptide loading and exchange catalyst[6,12–18]. These concerted actions ensure that only MHC I molecules loaded with optimal peptide epitopes are released from the PLC.

In addition to acquiring high-affinity peptides, glycosylation is immensely important for MHC I structure, stability, and quality control, and regulates pMHC I egress to the cell surface[1,4–6,19]. MHC I heavy chains (hc) are co-translationally N-linked glycosylated at a conserved asparagine (Asn86) with a branched $Glc_3Man_9GlcNAc_2$ glycan structure (Glc, glucose; Man, mannose; GlcNAc, N-acetylglucosamine). After trimming of the two outermost glucose residues by ER-resident glucosidases I (GluI) and II (GluII), mono-glucosylated MHC I molecules are specifically recognized by the lectin-like chaperone calnexin, which cooperates with the disulfide isomerase ERp57 in MHC I hc folding[1,14]. Upon binding of $\beta_2$-

[1]Institute of Biochemistry, Biocenter, Goethe University Frankfurt, Max-von-Laue-Str. 9, 60438 Frankfurt am Main, Germany. [2]Oxford Glycobiology Institute, Department of Biochemistry, University of Oxford, OX1 3QU Oxford, UK. [3]These authors contributed equally: Alexander Domnick, Christian Winter. ✉ e-mail: tampe@em.uni-frankfurt.de

microglobulin ($\beta_2$m) to the MHC I hc, calnexin is replaced by calreticulin, which recruits the mono-glucosylated MHC I to the PLC for peptide editing and quality control[1,6]. Removal of the final glucose of the $Glc_1Man_9GlcNAc_2$ glycan by the GluII heterodimer is a critical step in MHC I maturation and is obligatory for trafficking of MHC I to the cell surface via the secretory pathway[19–21]. However, the sequence of events during peptide editing and glycan processing, and the potential communication between these processes, has not yet been examined for MHC I or any other client-chaperone complexes in the endoplasmic reticulum quality control (ERQC) pathway.

Here, we reveal the molecular underpinnings of epitope proofreading and MHC I quality control within the PLC by structural and biochemical means. Although the overall architecture of the human PLC was reported recently[10], structural elements crucially involved in its catalytic mechanisms have remained unresolved due to the limited resolution. In this study, we determine the 3.7-Å single-particle cryogenic electron microscopy (cryo-EM) structure of the PLC editing module reconstituted in lipid nanodiscs (Nd), which enables us to identify pivotal features of this supramolecular chaperone assembly. Combining structural and functional studies, we discover that MHC I N-glycan processing is allosterically coupled to peptide editing, thus unraveling the central steps at the junction between pMHC I assembly and ERQC.

## Results

### MHC I molecules exhibit a homogeneous glycan structure in the PLC

To study the PLC in a near-native lipid environment, we isolated detergent-solubilized human PLC from Burkitt's lymphoma cells using the herpesviral protein ICP47 as bait[10]. After affinity purification, the PLC was reconstituted into large nanodiscs (-16 nm diameter) using the membrane scaffold protein MSP2N2. Size-exclusion chromatography (SEC) revealed a homogeneous population of PLC/MSP2N2/lipid particles (Fig. 1a). SDS-PAGE and immunoblotting showed identical stoichiometries of the reconstituted PLC subunits TAP1, TAP2, tapasin, ERp57, calreticulin, MHC I hc, $\beta_2$m, and the viral factor ICP47, when compared to detergent-solubilized PLC (Fig. 1b).

To determine the composition of N-linked glycans of the PLC glycoproteins, we analyzed the PLC by liquid chromatography-mass spectrometry (LC-MS) (Supplementary Fig. 1, 2, and Supplementary Table 1). Almost all MHC I molecules (>95%) carried a $Glc_1Man_9GlcNAc_2$ glycan, exemplified by a molecular weight (MW) of 40,340 Da for the most prevalent allomorph HLA-A*03:01 (Fig. 1c). PNGase F treatment led to a shift of 2027 Da (MW of $Glc_1Man_9GlcNAc_2$ minus water), consistent with the removal of the entire N-linked glycan (Fig. 1c). In contrast to MHC I, the glycan of tapasin linked to Asn233 was compositionally heterogeneous (Fig. 1d). While the main tapasin species carried a $Man_9GlcNAc_2$ glycan lacking the three terminal glucose

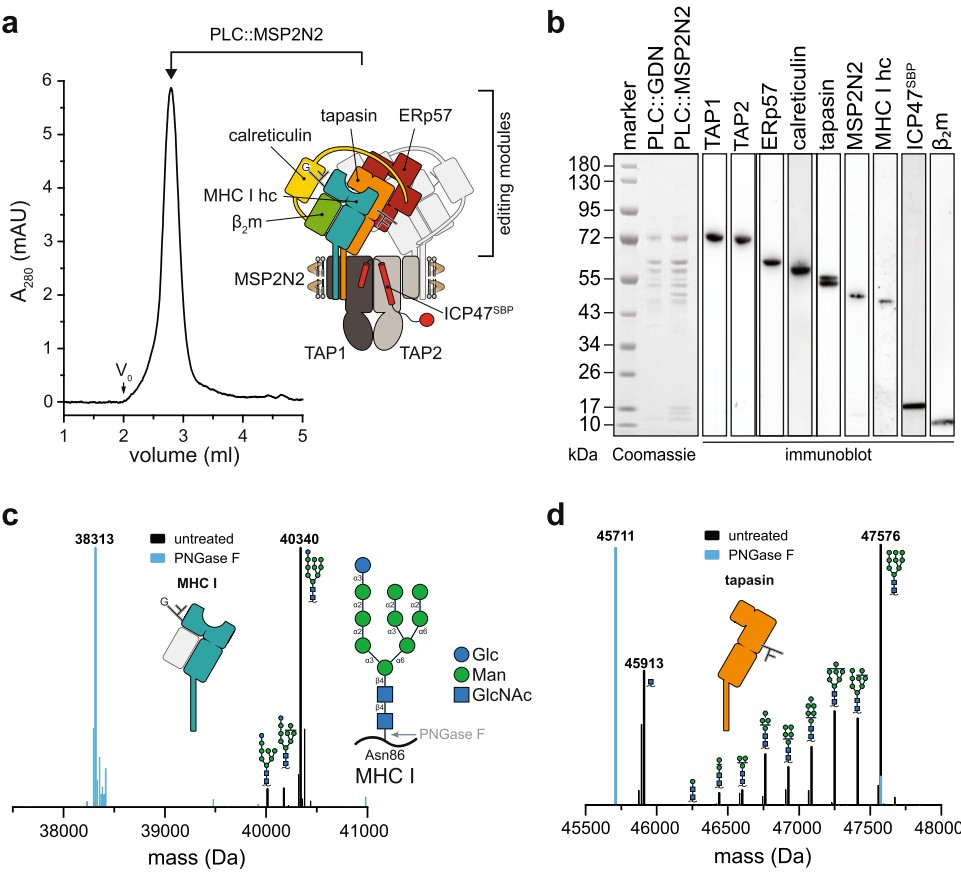

**Fig. 1 | PLC reconstituted in large lipid nanodiscs displays distinct glycosylation modification of MHC I and tapasin. a** PLC reconstituted in large lipid nanodiscs (PLC::MSP2N2) analyzed by size-exclusion chromatography (SEC). The PLC contains two editing modules, centered around the peptide transporter TAP1/2. For better visualization, one of the two editing modules is shown in light grey. **b** Compositional analysis of human PLC purified by ICP47[SBP] in detergent (PLC::GDN) and reconstituted in lipid nanodiscs (PLC::MSP2N2) by SDS-PAGE, immunoblotting, and LC-MS (Supplementary Fig. 1 and 2). **c** Glycosylation pattern of MHC I allomorph HLA-A*03:01 associated with the PLC ($Glc_1Man_{9-7}GlcNAc_2$), analyzed by mass spectrometry (deconvoluted spectrum, black). PNGase F-treated sample is shown as reference in cyan. HLA-A*03:01 is the predominant allomorph in LC-MS analysis. **d** Glycosylation of tapasin in the PLC ($Man_{9-0}GlcNAc_{2-1}$), analyzed by mass spectrometry (deconvoluted spectrum, black). PNGase F-treated sample is shown as reference in cyan. The data in this figure are representative of two biological replicas. Source data are provided as a Source Data file (**a**, **b**) or in the Zenodo open access repository (**c**, **d**).

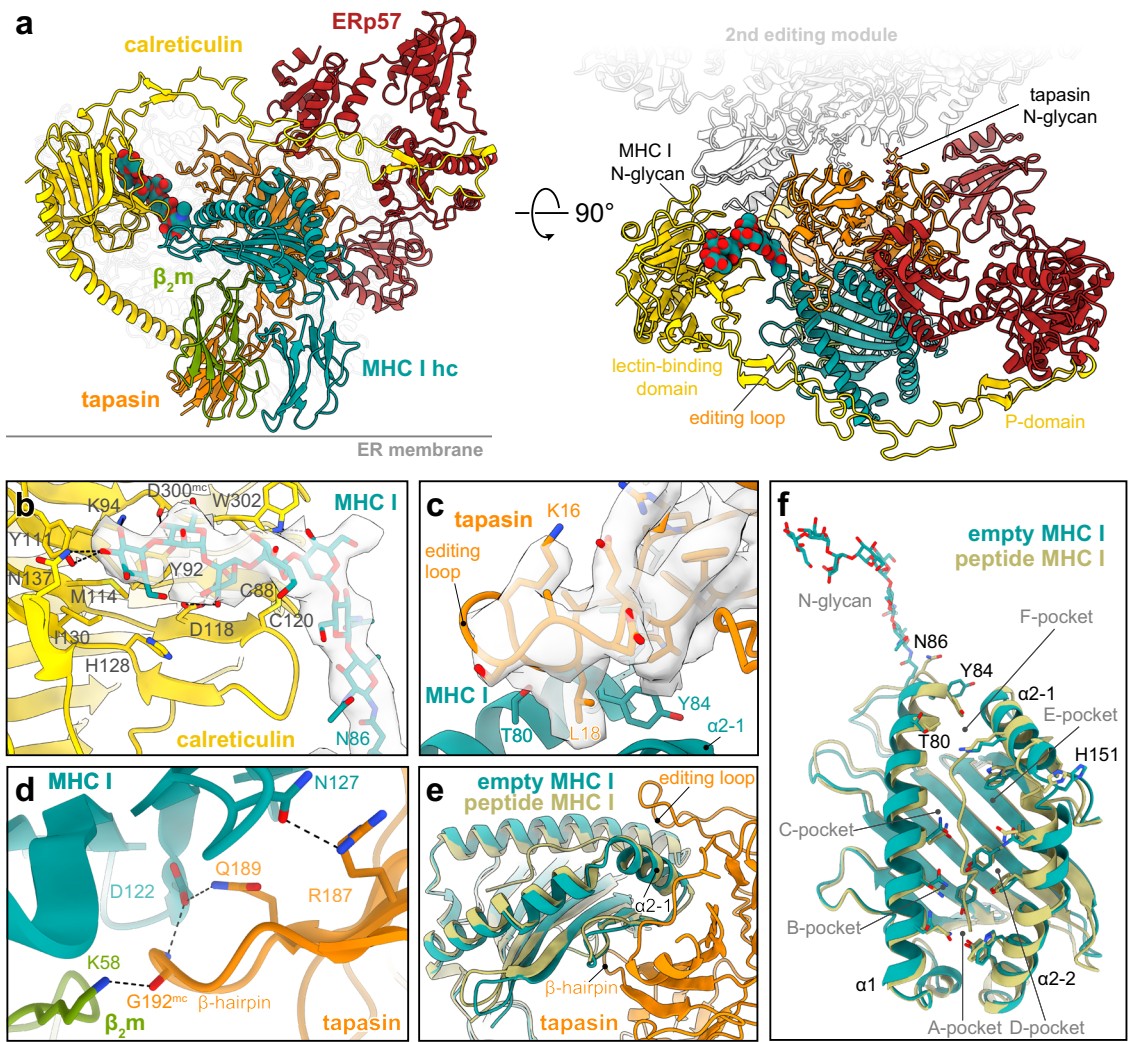

**Fig. 2 | Structure of the multivalent MHC I chaperone network in the PLC.**
**a** Cryo-EM structure of the peptide-receptive MHC I stabilized by a multivalent chaperone network in the PLC editing module (side and top view). The glycan trees of MHC I and tapasin are shown as space-filling model and stick model, respectively. Each subunit is colored separately ($\beta_2$m, green; calreticulin, yellow; ERp57, red; MHC I hc, teal; tapasin, orange). The position of the second editing module is indicated in light grey. **b** Interaction of the MHC I glycan with the lectin domain of calreticulin. The cryo-EM map of the glycan is depicted as a transparent isosurface (contour level: 0.19, light grey). **c** Tapasin editing loop and interactions with the MHC I hc molecule. The cryo-EM map of the editing loop is depicted as transparent isosurface (contour level: 0.19, light grey). **d** $\beta$-hairpin of tapasin and interactions with $\beta_2$m and MHC I hc. **e** Superposition of peptide-loaded MHC I (khaki, HLA-A*03:01, PDB ID: 3RL1) with the empty MHC I of the PLC (side view, including the interface with tapasin). **f** Superposition of PLC-associated peptide-receptive MHC I with peptide-loaded MHC I (khaki, HLA-A*03:01, PDB ID: 3RL1). Source data is available at EMDB (EMD-14119) and PDB (7QPD).

residues, we observed the full range of trimmed glycans (Man$_{8-0}$-GlcNAc$_2$), even down to a single GlcNAc as minimal N-glycan modification. This unusual mannose trimming indicates that tapasin goes through multiple rounds of glycan processing while being protected against ER-associated degradation (ERAD).

### Peptide-receptive MHC I are stabilized by a multi-chaperone network

To better understand the multi-chaperone network within reconstituted PLC, we determined the single-particle cryo-EM structure of its editing module to 3.7 Å resolution (Fig. 2). Reconstitution in lipid nanodiscs was crucial to stabilize the transient multi-chaperone machinery for cryo-EM analyses without chemical fixation[10]. As previously shown, 2D and 3D classifications revealed that the PLC is organized in two editing modules centered around the peptide transporter TAP1/2 (Supplementary Figs. 3–5). Despite the confinement in large lipid nanodiscs, the two editing modules and the transport complex TAP remained flexible relative to each other, limiting the

overall resolution of the entire PLC. 3D variation analysis depicts that one editing module was always fully assembled, showing tapasin, ERp57, calreticulin, MHC I hc, and $\beta_2$m, while the second one was displaced and lacked some of the PLC components (Supplementary Fig. 4). Focused refinement on the fully assembled editing module resulted in a cryo-EM map with an average resolution of 3.7 Å (Fig. 2a, Supplementary Fig. 5a–d and Supplementary Table 2).

As major interaction hub, tapasin contacts all other PLC subunits (Fig. 2a). In addition to the non-covalent interactions, tapasin is disulfide-linked via Cys95 to Cys33 of ERp57 (Supplementary Fig. 6a). The high-resolution structure allowed us to build an atomic model of the N-glycan bridging the lectin domain of calreticulin to Asn86 of MHC I hc (Fig. 2b). The terminal glucose is coordinated by Lys94, Asn137, and Tyr111 of calreticulin, consistent with the X-ray structure of the isolated lectin domain bound to a Glc$_1$Man$_3$ tetrasaccharide[22]. Whereas Met114 and Ile130 of calreticulin shape the glucose-binding pocket, further contacts to the N-glycan are established by Asp118, Asp300$^{mc}$ (main chain), Tyr92, and Trp302. Additional calreticulin

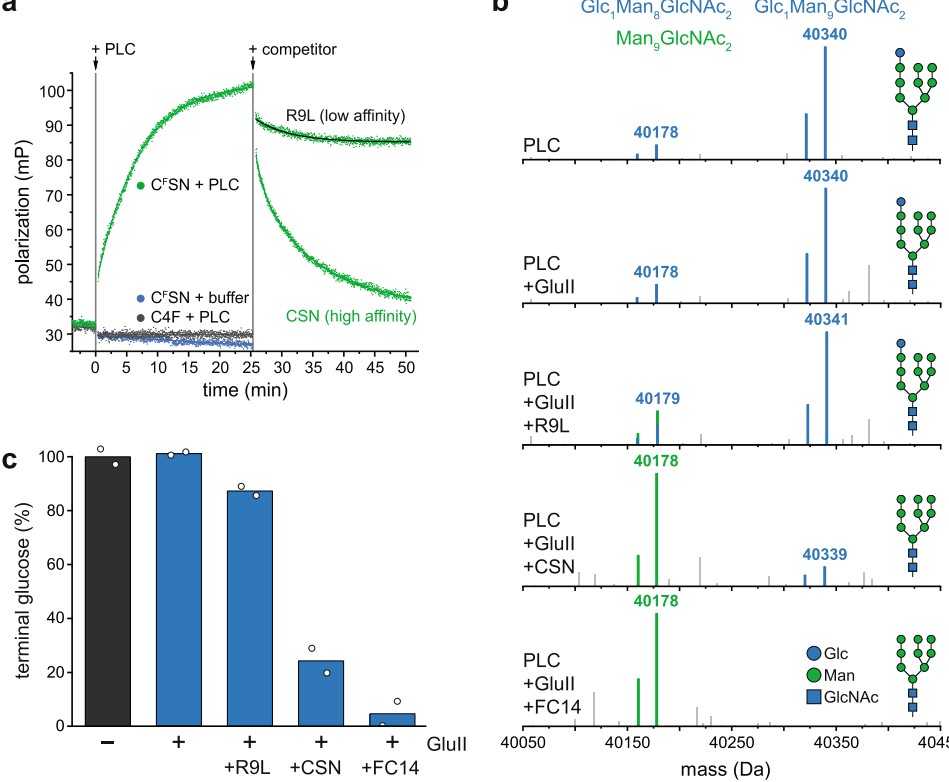

**Fig. 3 | Peptide editing and allosterically coupled MHC I glycan processing by GluII in the PLC. a** Peptide binding and proofreading of HLA-A*03:01 associated with the PLC reconstituted in lipid nanodiscs (75 nM of PLC) monitored by fluorescence polarization using equimolar concentrations of fluorescent reporter peptides: high-affinity epitope C$^F$SN (AIFC$^F$SNMTK) or low-affinity HLA-B*27:01-restricted epitope C4F (RRYC$^F$KSTEL). For peptide editing, a 2000-fold molar excess of unlabeled high-affinity peptide (150 μM CSN, AIFCSNMTK) or low-affinity peptide (150 μM R9L, RRYQKSTEL, black trace) were used. **b** LC-MS analysis of HLA-A*03:01 glycan status during glycan processing. **c** The percentage of the Glc$_1$Man$_9$GlcNAc$_2$-modified HLA-A*03:01 is given as mean ($n$ = 2). The data are representative of two biological replicas. Source data are provided as a Source Data file (**a**) or in the Zenodo open access repository (**b, c**).

residues involved in glycan binding are His128 and the disulfide bond between Cys88 and Cys120 (Fig. 2b). The mannoses in the B/C branch of the glycan were too flexible to be resolved at high resolution. In contrast to the N-glycan of MHC I, only the trisaccharide Man$_1$GlcNAc$_2$ stem of the Asn233-linked glycan in tapasin was defined by the cryo-EM map, suggesting that the remaining sugar moieties of the tapasin-linked glycan are flexible.

Besides the structurally defined N-linked glycan of MHC I, hereafter referred to simply as "MHC I glycan", our reconstruction revealed crucial features of the tapasin-MHC I multi-chaperone complex which have not been elucidated so far. A well-resolved loop of tapasin (residues 11–20), henceforth referred to as editing loop, is positioned on top of the F-pocket of the empty MHC I peptide-binding groove (Fig. 2c, Supplementary Fig. 6b). Leu18 of the editing loop contacts MHC I in a position that would clash with Tyr84 of peptide-bound MHC I (Fig. 2c). Thus, the structure suggests that Leu18 disrupts the contact of Tyr84 with the C terminus of cargo peptides and thereby contributes to peptide exchange catalysis[23]. This arrangement resembles the position of the scoop loop in TAPBPR, which is larger (16 amino acids vs. 10 in tapasin) but is very similarly positioned and also disturbs the interaction between Tyr84 in MHC I and the C terminus of the cargo peptide by inserting residues 34–36 into the MHC I peptide binding groove[24]. This creates a larger interface with the binding groove than the tapasin loop[25]. The displacement of Tyr84 is stabilized by the interaction with Glu72 of tapasin, consistent with TAPBPR-MHC I crystal structures[24,26]. Leu18 of the editing loop also supports the open MHC I conformation by pushing away Thr80 and Leu81 in the α1 helix, as well as Ala139 and Ile142 in the α2-1 helix of MHC I. HLA-A*03:01, the predominant allomorph in the isolated PLC,

harbors an acidic F-pocket, and it was hypothesized that this acidic F-pocket is stabilized by Lys16 of the editing loop[23]. However, this is not what we observed. On the contrary, the side chain of Lys16 is flexible in our structure and points away from the MHC I peptide-binding groove, which is consistent with previous simulations[25]. Notably, rat and mouse tapasin do not contain any basic residue in the editing loop, rendering the proposed F-pocket stabilization as a general accessory catalytic principle unlikely. Consistent with the structure of TAPBPR-MHC I chaperone complexes[24,26], the floor of the peptide-binding groove is acted upon by a β-hairpin of tapasin (Fig. 2d). Direct contacts in this region include Arg187 and Gln189 of tapasin, as well as Asn127 and Asp122 of MHC I. The latter residue also forms a hydrogen bond with the main-chain of Gly192 at the tip of the tapasin β-hairpin, which additionally interacts with Lys58 of β$_2$m. The contact of Arg187 in tapasin with the MHC I hc might contribute to substrate specificity of the peptide editor, as the interacting MHC I residue (Asn127 in HLA-A*03:01) is not conserved between allomorphs.

A superimposition of the chaperone-stabilized empty MHC I onto the X-ray structure of soluble, non-glycosylated peptide-loaded HLA-A*03:01 (ref. 27) highlights the open conformation of PLC-bound MHC I (Fig. 2e, f). The rearrangement of the α1 and α2 helices results in a widening of the peptide-binding groove, facilitating peptide exchange. The α1 helix (residue 57–85) displays Cα shifts which are most pronounced at the F-pocket (residues 79–85), with a root mean square deviation (RMSD) of 1.8 Å. Interaction with tapasin leads to a lateral shift of the α2 helix which is most pronounced in the hinge region (residues 149–153), with an RMSD of 2.0 Å. In contrast to TAPBPR-MHC I complexes[6,24,26], the empty peptide-binding groove of the PLC-bound MHC I is in its widened conformation stabilized on

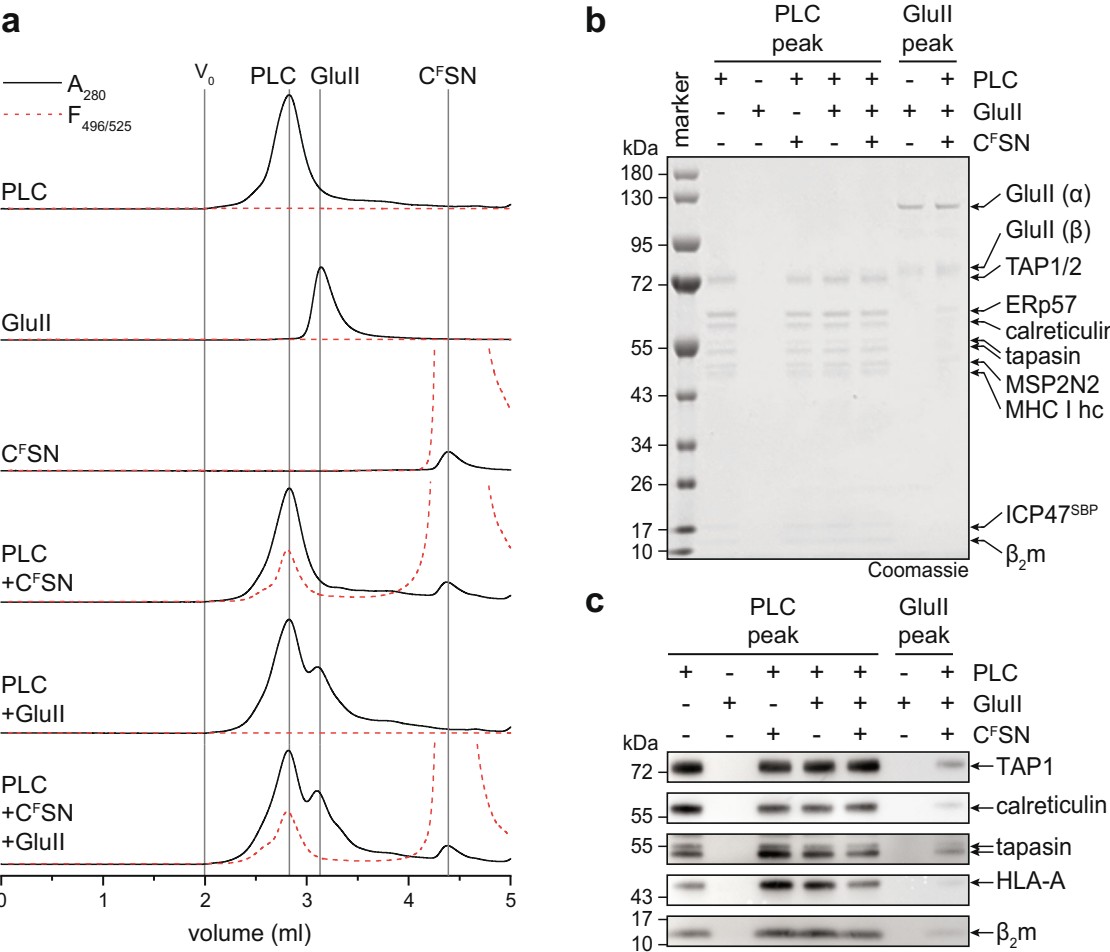

**Fig. 4 | PLC remains fully assembled after peptide editing and glycan trimming by GluII. a** The size of the Nd-reconstituted PLC remains unchanged after peptide binding and GluII trimming as demonstrated by SEC. Peptide loading and glycan trimming by GluII does not change the overall composition and stoichiometry of the PLC. **b**, **c** The PLC remains fully assembled after MHC I loading with high-affinity peptide epitope (CSN) and GluII-catalyzed deglucosylation of MHC I, as demonstrated by SDS-PAGE (**b**) and immunoblotting (**c**) of SEC fractions corresponding to the PLC and GluII peak. The data are representative of two biological replicas. Source data are provided as a Source Data file.

both helices, not only by multivalent protein-protein interactions with tapasin, but also by glycan-protein contacts via calreticulin. Based on the shift of the α1 helix, we hypothesized that the closing of the peptide-binding groove after successful peptide editing might result in a release of the glycan from the lectin domain of calreticulin.

### The PLC reconstituted in near-native lipid environment is functional in peptide proofreading

Peptide proofreading by tapasin ensures that only kinetically stable pMHC I loaded with high-affinity peptides are released to the cell surface, whereas suboptimal, low-affinity peptides are discharged and eventually replaced by high-affinity epitopes[6,12–15,17,18]. Once MHC I molecules are loaded with an optimal epitope, they must leave the PLC to present the peptide cargo on the cell surface. To pass ERQC, the terminal glucose of the Asn86-linked MHC I glycan must be removed by GluII. We analyzed the peptide proofreading activity of lipid-embedded PLCs by fluorescence polarization using the fluorescein ($^F$)-labeled peptide C$^F$SN (AIFC$^F$SNMTK). This HIV-Nef73-derived epitope binds with high affinity to HLA-A*03:01, the predominant MHC I allomorph in PLCs isolated from Burkitt's lymphoma cells. Within 15 min, binding of C$^F$SN was observed upon its addition to PLC-bound MHC I (Fig. 3a). In contrast, the suboptimal peptide C4F (RRYC$^F$KSTEL) did not interact with the PLC. Importantly, bound C$^F$SN peptides were rapidly displaced by an

excess of unlabeled CSN peptide, while a similar excess of the unfavored peptide R9L (RRYQKSTEL) did not trigger substantial peptide exchange (Fig. 3a). Thus, the reconstituted PLC was fully functional in catalyzing peptide proofreading.

### Glycan processing is allosterically coupled to peptide editing

To monitor the glycosylation status of HLA-A*03:01 during peptide proofreading, we established an in-vitro glycan processing assay. Reconstituted PLCs were subjected to GluII trimming and analyzed by LC-MS in the presence and absence of the high-affinity peptide CSN or in the presence of the non-binding peptide R9L (Fig. 3b, c). In the absence of peptides, the addition of GluII did not change the glycan of MHC I ($Glc_1Man_9GlcNAc_2$, 40,340 Da), demonstrating that the terminal glucose is protected from processing by GluII (Fig. 3b, c). However, in the presence of the high-affinity peptide CSN, a single glucose moiety was removed by GluII, reflected by a mass shift of 162 Da ($Man_9$-$GlcNAc_2$-MHC I, 40,178 Da) (Fig. 3b, c, Supplementary Fig. 7). In contrast, processing of the MHC I glycan by GluII was not induced by addition of the unfavored peptide R9L (Fig. 3b, c). Notably, the MHC I glycan was completely deglucosylated after disassembly of the PLC by a short incubation with the detergent Fos-choline-14 (Fig. 3b, c). These observations indicate that GluII-mediated glycosidic bond cleavage of the innermost glucose residue on the MHC I glycan is allosterically coupled to peptide editing.

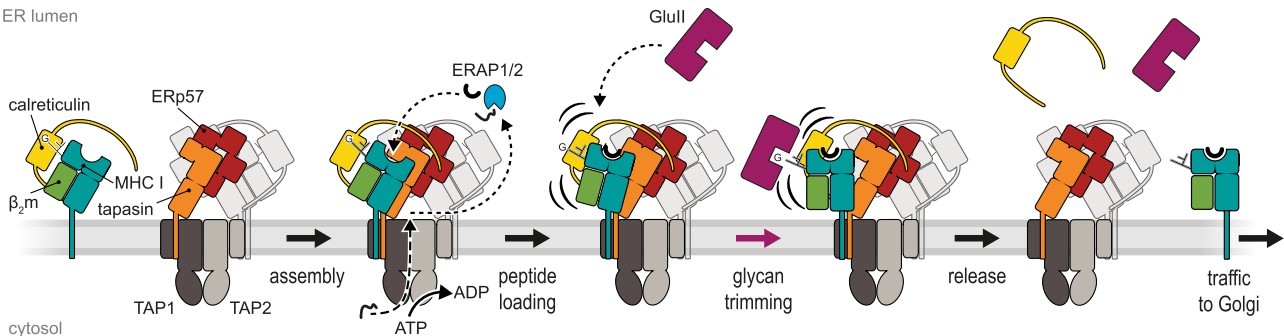

**Fig. 5 | Mechanism of MHC I assembly, peptide editing, and quality control.**
Peptide-receptive MHC I heterodimers are recruited by calreticulin to the PLC and form the fully assembled PLC, which is composed of two editing modules. The peptide transporter TAP1/2 shuttles peptides from the cytosol into a molecular basket formed by the editing modules. Two lateral windows allow peptides to diffuse into the ER lumen to be edited by the ER-resident aminopeptidase ERAP1/2 and subsequently loaded onto MHC I molecules. After tapasin-facilitated peptide loading and proofreading of pMHC I, the MHC I glycan becomes accessible for trimming by GluII in the presence of calreticulin. Thus, calreticulin is not required to leave the PLC for N-glycan editing when MHC I release is restricted. Glucose-trimmed pMHC I complexes are released from the PLC and traffic via the Golgi compartment to the cell surface. The remaining asymmetric PLC resides in the membrane awaiting the next calreticulin-associated MHC I heterodimers.

## Glycan trimming and peptide proofreading are coordinated within the PLC

Based on the allosteric coupling of peptide proofreading and MHC I glycan processing, we finally addressed the question whether calreticulin must leave the PLC for steric reasons to allow glycan trimming by the GluII heterodimer, as suggested by current working models of ERQC[5,28,29]. To this end, we analyzed the PLC composition before and after peptide loading and glycan trimming. Using the high-affinity fluorescent peptide C$^F$SN as reporter, we followed peptide loading of MHC I in parallel by fluorescence-detection SEC. We did not observe any change in size and overall composition of the PLC upon peptide loading of MHC I and glycan trimming by GluII (Fig. 4), indicating that MHC I glycan processing by GluII occurs upon loading of optimal epitopes in the fully assembled PLC containing calreticulin (Fig. 4b, c). This implies that the supramolecular organization of the PLC is characterized by a significant degree of plasticity. Taken together, the finding of an allosteric coupling between peptide editing and glycan processing during MHC I quality control suggests that GluII is a transient component of the PLC providing additional layers of complexity in the organization and function of the PLC.

## Discussion

MHC I quality control is of fundamental importance for adaptive immunity. Here we provide structural and biochemical evidence for an allosteric coupling between glycan processing and peptide proofreading coordinated within the PLC. The high resolution of our cryo-EM reconstruction allowed us to build a precise model of the multi-chaperone-client complex including tapasin, ERp57, calreticulin, MHC I hc, and β$_2$m and to define key mechanistic features. The structure identifies the editing loop of tapasin and the calreticulin-engulfed MHC I glycan as crucial elements during MHC I quality control. Our glycan processing studies demonstrated that, as far as the Nd-reconstituted PLC is concerned, the status of the MHC I glycan is coupled to the peptide-loading status. Based on our structural data and the identification of context-sensitive MHC I glycan processing, we propose the following model for peptide loading and quality control of MHC I (Fig. 5): After recruitment of peptide-receptive MHC I in complex with calreticulin into the PLC, MHC I complexes acquire high-affinity peptide epitopes through tapasin-catalyzed proofreading and peptide exchange. Binding of an optimal epitope weakens the interaction between MHC I and tapasin, but also leads to a rearrangement of its MHC I glycan. This rearrangement loosens the connection between the A branch of the glycan and the lectin domain of calreticulin and allows deglucosylation by GluII as a prerequisite for pMHC I

release from the PLC and subsequent trafficking to the cell surface. This model entails that peptide loading and proofreading of MHC I molecules are coupled to glycan trimming by GluII in a fully assembled PLC, establishing the role of the MHC I glycan as allosteric sensor in quality control. A limitation of the study is that the diffusion of membrane-bound PLC components, including the client (p)MHC I, was restricted by the membrane scaffold. Consequently, MHC I molecules are always in proximity to the PLC, and we could not determine whether pMHC I leaves the PLC before or after GluII trimming. For future experiments, it would be necessary to study the PLC in continuous membranes that allow free diffusion of the PLC and its components.

The structural and mechanistic data also provide the molecular basis of the specificity for glycosylated MHC I in the case of the PLC as opposed to the glycan-independent TAPBPR-MHC I interaction[30]. The observation that GluII-mediated glycan trimming of pMHC I takes place within the calreticulin-containing PLC establishes the notion that the PLC is a highly dynamic and malleable system able to accommodate large transient components. Thus, the PLC emerges as a supramolecular assembly that is more complex than previously anticipated, and which might function as an interaction hub for additional ERQC factors beyond GluII. Our study illustrates the core principles of how sophisticated multi-chaperone systems involving networks of both protein-protein and protein-glycan interactions orchestrate folding, assembly, and quality control of a client in the ER. In the case of MHC I, the multi-chaperone machinery ensures that only stable pMHC I complexes with optimal peptide epitopes are presented at the cell surface, thereby allowing the adaptive immune system to mount an efficient response against pathogens and cancer while avoiding autoimmune reactions. We expect that the prototypical example of tightly coordinated and dynamic quality control by the PLC provided here can be extended to ERQC processes in general.

## Methods

### ICP47$^{SBP}$ preparation

The ICP47$^{SBP}$ sequence was cloned into a pETM-11 (European Molecular Biology Laboratory, EMBL) using the NcoI and BamHI restriction sites. *Escherichia coli* One Shot BL21(DE3) cells (Thermo Fisher) were transformed and grown to OD$_{600}$ = 0.6 at 37 °C. Expression was induced with 0.2 mM isopropyl-β-D-thiogalactoside (IPTG) for 6 h at 22 °C. The cells were lysed by sonication in 50 mM NaP$_i$ pH 8.0, 150 mM NaCl, 20 mM imidazole, 0.2% Tween 20, 2 mM DTT, supplemented with 2.5 mM PMSF, 3.75 mg/mL lysozyme, 6.25 mM benzamidine, and 1 U/mL benzonase. His$_6$-TEV-ICP47$^{SBP}$ was isolated from the lysate via reverse IMAC (Ni-NTA Agarose resin, QIAGEN). After washing with

50 mM NaP$_i$ pH 8.0, 150 mM NaCl, 20 mM imidazole, 0.2% Tween 20, 10 mM DTT, ICP47$^{SBP}$ was eluted supplementing the wash buffer with 500 mM imidazole. The His$_6$-tag was removed by overnight TEV protease digestion. ICP47$^{SBP}$ was purified by reversed phase C$_{18}$ HPLC (Agilent, 1200 Series System; PerfectSil 300 ODS C$_{18}$ 5 µm 300x10 mm) applying a linear water/acetonitrile gradient from 5–60% supplemented with 0.1% (v/v) TFA. The identity of ICP47$^{SBP}$ was confirmed by LC-MS analysis (M$_{calc}$: 14694.06 Da, M$_{obs}$: 14694.75 Da).

## PLC purification

The native PLC was isolated from Burkitt's lymphoma cells (Raji ATCC® CCL-86), cultured in RPMI 1640 medium (Gibco), supplemented with 10% Fetal Calf Serum (FCS, Capricorn), 3 mM HEPES-NaOH pH 7.5 (Gibco) at 37 °C and 8% CO$_2$ in a shaking incubator (Eppendorf). Cells were harvested by centrifugation, snap frozen in liquid nitrogen, and stored at −80 °C until further use. Cell pellets were thawed and resuspended in 20 mM HEPES-NaOH pH 7.4, 150 mM NaCl, 10 mM MgCl$_2$, protease inhibitor mix (Serva) and incubated with ICP47$^{SBP}$ for 15 min at 4 °C. Membranes were mixed with 2% (w/v) glyco-diosgenin (GDN, Anatrace) by douncing and incubated for 2 h at 4 °C under agitation. Insoluble material was removed by centrifugation (45 min, 100,000 × g). ICP47$^{SBP}$-arrested PLC was bound to Streptavidin High-Capacity Agarose (Pierce) and washed extensively. The PLC was either directly eluted in 20 mM HEPES-NaOH pH 7.4, 150 mM NaCl, 0.05% (w/v) GDN, 2.5 mM biotin (PLC::GDN) or reconstituted into MSP2N2 nanodiscs on the beads (PLC::MSP2N2).

## MSP2N2 preparation

pMSP2N2 was a gift from Stephen Sligar (Addgene plasmid #29520). Membrane scaffold protein MSP2N2 was expressed and purified as described[31]. In brief, MSP2N2 was expressed in *E. coli* BL21(DE3) grown in LB media supplemented with 0.5% (w/v) glucose at 37 °C. At an OD$_{600}$ = 0.8, expression was induced with 1 mM IPTG, and cells were grown for 1 h at 37 °C. Subsequently, the temperature was lowered to 28 °C, and cells were grown for an additional 4 h. Cells were disrupted by sonication in lysis buffer (50 mM Tris-HCl pH 8.0, 500 mM NaCl, 1% (w/v) Triton X-100, 4% Na-deoxycholate, protease inhibitor mix (Serva)). The protein was purified via its N-terminal His$_7$-tag.

## PLC reconstitution in membrane scaffolds

PLC bound to Streptavidin High Capacity Agarose (Pierce) was reconstituted in MSP2N2 nanodiscs in a molar ratio of 1:40:500 (PLC:MSP2N2:lipids). First, PLC was incubated with bovine brain lipids (Sigma) for 15 min and 4 °C under constant agitation in buffer with reduced detergent concentration (20 mM HEPES-NaOH pH 7.4, 150 mM NaCl, 0.003% (w/v) GDN). Afterwards, MSP2N2 was added and incubated for further 30 min at 4 °C. To induce nanodisc formation, Biobeads SM-2 (Bio-Rad) were added and incubated for 1 h at 4 °C under constant agitation. Another batch of Biobeads was added for 1 h, 4 °C. The agarose resin was washed extensively with buffer without detergent (20 mM HEPES-NaOH pH 7.4, 150 mM NaCl) and reconstituted PLC::MSP2N2 eluted by biotin (20 mM HEPES-NaOH pH 7.4, 150 mM NaCl, 2.5 mM biotin). For subsequent experiments, PLC::MSP2N2 was concentrated using a 100 kDa MWCO filter concentrator (Amicon).

## Biochemical analysis of the PLC

For SDS-PAGE, precast NuPAGE gradient gels (Novex) were used. Gels were either stained by Instant Blue (Expedeon) or directly transferred to a PVDF membrane (Bio-Rad). PLC composition was verified by using the antibodies, anti-TAP1 (mAb 148.3, hybridoma supernatant dilution 1:20)[32], anti-TAP2 (mAb 438.3, hybridoma supernatant dilution 1:20)[33], anti-tapasin (Abcam, catalogue number ab13518, dilution 1:1000), anti-HLA-A/B/C HC10 (Acris Antibodies, catalogue number AM33035PU-N, dilution 1:1000), anti-HLA-A (Abcam, catalogue number ab52922,

dilution 1:1000), anti-ERp57 (Abcam, catalogue number ab10287, dilution 1:2000), anti-calreticulin (Sigma, catalogue number C4606, dilution 1:1000), and anti-β$_2$m (Novo Antibodies, catalogue number HPA006361, dilution 1:1000). ICP47$^{SBP}$ was detected by the anti-SBP antibody (Santa Cruz, catalogue number sc-101595, dilution 1:500), MSP2N2-His$_6$ was detected with the anti-His$_6$ antibody (Sigma, catalogue number H1029, dilution 1:1000). Integrity of the PLC was verified by size-exclusion chromatography (SEC). SEC analysis was performed at 4 °C using a Shimadzu HPLC system, equipped with a Shodex semi-micro KW404-4F (4.6 × 300 mm) column. A running buffer containing 20 mM HEPES-NaOH pH 7.4, 150 mM NaCl was used for PLC::MSP2N2 samples. For PLC::GDN samples, the buffer was supplemented with 0.01% (w/v) GDN. Uncropped gels and immunoblots are provided within the source data file.

## Peptide design, labeling, and purification

The high-affinity HLA-A*03:01 binder AIFQSSMTK ($K_D \approx 10$ nM, HIV-derived epitope) was chosen to create the CSN peptide (AIFCSNMTK, $K_D \approx 10$ nM) for fluorescent labeling. Peptide affinities were predicted with NetMHC 4.1 (ref. 34). Peptides were synthesized by automated microwave-assisted solid-phase peptide synthesis by Fmoc chemistry (CEM, Liberty Blue). The CSN peptide and C4 peptide (RRYCKSTEL, $K_D \approx 20$ µM) were labeled with 5-iodoacetamide fluorescein (5-IAF, Sigma–Aldrich) in PBS/DMF buffer (8.1 mM Na$_2$HPO$_4$ pH 6.5, 137 mM NaCl, 2.7 mM KCl, 1.8 mM KH$_2$PO$_4$, 33% (v/v) DMF) with heavy agitation for 2 h at 20 °C using a two-fold molar excess of 5-IAF. Samples were purified by reversed phase C$_{18}$ HPLC (Agilent, 1200 Series System; PerfectSil 300 ODS C$_{18}$ 5 µm 300 × 10 mm) applying a linear water/acetonitrile gradient from 5–60% supplemented with 0.1% (v/v) TFA. Purified peptides were snap frozen in liquid nitrogen and lyophilized (Lyovac GT2, Heraeus). Peptide identity was confirmed by LC-MS analysis (AIFCSNMTK: [M + H$^+$]$_{calc}$: 1014.475 Da, [M + H$^+$]$_{obs}$: 1014.475 Da; AIFC$^F$SNMTK: [M + H$^+$]$_{calc}$: 1401.550 Da, [M + H$^+$]$_{obs}$: 1401.549 Da; RRYQKSTEL: [M + H$^+$]$_{calc}$: 1180.636 Da, [M + H$^+$]$_{obs}$: 1180.635 Da; RRYC$^F$KSTEL: [M + H$^+$]$_{calc}$: 1542.669 Da, [M + H$^+$]$_{obs}$: 1542.663 Da).

## Peptide binding and editing in the PLC

Binding of fluorescein ($^F$)-labeled peptides C$^F$SN (AIFC$^F$SNMTK) or C4F (RRYC$^F$KSTEL) was followed by fluorescence polarization. The polarization of fluorescent peptides (75 nM) in buffer was analyzed at $\lambda_{ex/em}$ of 485/520 nm using a microplate reader (CLARIOstar, BMG LABTECH). Afterwards, PLC::MSP2N2 was added to a final concentration of 75 nM, and the sample was mixed 5 s before polarization recording (500 rpm, double orbital mode). 150 µM of CSN (AIFCSNMTK) or R9L (RRYQKSTEL) peptide was added for competition and the measurement continued. The fluorescence anisotropy was calculated using:

$$r = \frac{I_{\parallel} - I_{\perp}}{I_{\parallel} + 2 \times I_{\perp}} \tag{1}$$

Resulting curves were fitted using OriginPro 2020 by using linear fit for constant data (fitting function: $y = y_0 + b*x$), exponential fitting with either one factor for binding and R9L competition (fitting function: $y = y_0 + A*\exp(k*x)$) or two factors for CSN competition (fitting function: $y = y_0 + A_1*\exp(-k_1 x) + A_2*\exp(-k_2 x)$).

## LC-MS analysis

All LC-MS measurements were performed with a BioAccord System (Waters) running Unify 1.9.4 (Waters). Peptides were analyzed with an ACQUITY UPLC Peptide BEH C$_{18}$ Column, 130 Å, 1.7 µm, 2.1 mm × 100 mm (Waters), applying a linear water/acetonitrile gradient supplemented with 0.1% (v/v) formic acid at 60 °C, 30 V cone voltage, 0.8 kV capillary voltage, and 550 °C desolvation temperature. Mass spectra were recorded in positive polarity at 5 Hz in MS$^e$ mode at 50–2000 m/z. Intact protein LC-MS measurements were acquired on an ACQUITY

UPLC Protein BEH $C_4$ Column, 300 Å, 1.7 μm, 2.1 mm × 150 mm (Waters) at 80 °C using a cone voltage of 60 V, 1.5 kV capillary voltage and 500 °C desolvation temperature. Mass spectra were recorded in positive polarity at 2 Hz in full scan mode at 400–7000 m/z. Masses of peptides and proteins were calculated and confirmed in Unify 1.9.4.053 (Waters). Intact mass spectra were deconvoluted in Unify using the integrated MaxEnt1 algorithm iterating to convergence. Spectra with high background noise were subjected to automatic baseline correction before deconvolution. Deconvoluted spectra were centroided based on peak height, and mock spectra were extracted. Centroided spectra were used for mass calculations. All intact protein mass spectra show the top 95% of signal intensity. UV spectra were recorded at 280 nm with 10 Hz.

### Deglycosylation by PNGase F

20 μL 0.5 mg/ml PLC in 20 mM HEPES-NaOH pH 7.4, 150 mM NaCl, 0.01% (w/v) GDN were heat-disintegrated for 15 min at 65 °C at 600 rpm. Subsequently, samples were incubated with 2 μL of PNGase F (NEB, 500 units/μL) at 37 °C. After 2 h, 2 μL of PNGase F were added. Samples were incubated overnight and directly subjected to LC-MS analysis.

### GluII expression and purification

The *Mus musculus GANAB* (α-GluII α-subunit, UniProt accession no.: Q8BHN3-2 with the amino acid substitution F724G) and *M. musculus PRKCSH* (α-GluII β-subunit, UniProt accession no.: O08795 with the amino acid substitutions L88P and S90N) were cloned into the mammalian expression vector pHLsec carrying a C-terminal $His_6$- or an N-terminal FLAG-tag, respectively. Co-transfection into the FreeStyle 293 Expression System (Life Technologies) took place following the manufactures protocol. Cells were maintained for four days at 37 ˚C, 8% $CO_2$, shaking at 135 rpm. Cells were pelleted by centrifugation at 3000 × *g* for 45 min. The supernatant was adjusted to 1x PBS (pH 7.4) and sterile filtered. The whole sample was applied to a HisTrap excel column (GE LifeSciences) pre-equilibrated with 1x PBS and subsequently washed with 5 mM imidazole, 1x PBS. Elution took place using 10 column volumes of 350 mM imidazole, 5% (w/v) glycerol, 1x PBS. The imidazole containing buffer of the eluate was exchanged with 5% (w/v) glycerol, 1x PBS using Ultra-15 spin concentrator (Amicon, 30 kDa MWCO) and the concentrated sample applied to a Superdex 200 16/600 column (GE Lifesciences) pre-equilibrated with 20 mM Hepes (pH 7.5) and 150 mM NaCl.

### Deglucosylation by GluII

1.2 μM of PLC::MSP2N2 in 20 mM HEPES-NaOH pH 7.4, 150 mM NaCl were preincubated with 20 μM of either high-affinity peptide (CSN), low-affinity peptide (R9L), or 1 mM FC14 for 30 min at 4 °C. Deglycosylation by GluII (0.6 μM final) was studied at 37 °C for 5 min. The reaction was stopped by 80 °C heat inactivation, and samples were analyzed by LC-MS.

### Cryo-EM sample preparation and data collection

3 μL of PLC::MSP2N2 (1.8 mg/ml, 2.5 μM) sample were applied onto freshly glow-discharged copper grids (Quantifoil, Cu R1.2/1.3) and plunge-frozen in liquid ethane using a Vitrobot Mark IV (Thermo Fisher). Micrographs were recorded automatically (SerialEM) on a 300-kV FEI Titan Krios in energy-filtered transmission electron microscopy mode with a K2 direct detector (Gatan) and a Gatan GIF Quantum SE post-column energy filter at 130,000x magnification and a pixel size of 1.05 Å. Dose-fractionated movies were acquired at an electron flux of 5.1 e⁻ per pixel per s over 14 s with 0.35 s exposures per frame (40 frames in total), corresponding to a total electron dose of ~65 e⁻ Å². Movies were recorded in the defocus range from −1.5 to −2.5 μm (Supplementary Table 2).

### Cryo-EM data analysis

Cryo-EM data analysis was performed using CryoSPARC v2.15–3.2 (ref. 35). A total of 2341 movies were used for analysis. Motion correction was performed using patch motion correction implemented in CryoSPARC. The contrast transfer function (CTF) was estimated using patch CTF function of CryoSPARC. Initial particles were picked from a subset of data using a blob picker (minimum particle diameter 150 Å, maximum particle diameter 400 Å). The particles were extracted at a box size of 360 px and subjected to 2D classification to identify good particle picks. These picks were used to train Topaz (deep picker) on denoised images[36,37]. Subsequently, the trained picking model was utilized to pick particles in all images, yielding 613,746 particles picked and extracted at a box size of 360 px. 2D classification was used to identify good particles for ab initio model generation, which directly yielded maps for full PLC including the membrane region as well as a more focused map consisting of one editing module only. These maps were used as references to identify suitable particles for refinement in several rounds of 3D classification. The 3D classification resulted in two clean particle stacks for the full PLC (52,668 particles) and the editing module (97,952 particles). Each subset of particles was individually refined by homogeneous and non-uniform refinement. Optimal results were obtained using non-uniform refinement utilizing a mask for the region of interest, enforced non-negativity, optimization of per particle defocus, and optimized per group CTF parameters enabled. For the focused editing submodule map, a resolution of 3.73 Å was estimated using the 0.143 cut-off criteria (Supplementary Fig. 5). For the full PLC, the highest resolution obtained was 4.01 Å. However, this map seemed to represent a consensus map of different PLC compositions of mainly one well resolved editing module; the second tapasin and MHC I were less represented. Separating the underlying states using ab initio model generation and/or heterogeneous refinement failed. Heterogeneous reconstruction with the same set of particles by deep learning using the neural network-based algorithm of CryoDRGN[38] (https://github.com/zhonge/cryodrgn) (version 0.3.1) revealed different assembly states of the PLC. CryoDRGN was run for 50 epochs with input image poses and CTF parameters from a consensus homogeneous reconstruction in cryoSPARC, encoder and decoder network architectures of 3 hidden layers with 256 nodes per layer, 8-dimensional latent space, and pose refinement. Selected maps generated by cryoDRGN representative of the latent space, as judged from a principal component analysis (PCA) projection, were used as references in heterogeneous 3D refinement in CryoSPARC and resulted in five subsets of particles representing different assembly states of the PLC (Supplementary Fig. 4). Individual stacks were subjected to homogeneous and non-uniform refinements resulting in resolutions between 6 Å and 9 Å.

### Model building

The structure of a PLC editing module (PDB ID: 6ENY) was initially fitted into the cryo-EM map using bulk flexible fitting by ISOLDE[39] implemented for ChimeraX. Final model building was carried out in COOT[40], and real-space refinement was performed using Phenix[41]. The editing loop of tapasin was modeled de novo. Carbohydrates were validated using Privateer[42] of the CCP4 software suite[43]. Refinement and validation statistics are summarized in Supplementary Table 2.

### Reporting summary

Further information on research design is available in the Nature Research Reporting Summary linked to this article.

## Data availability

LC-MS raw data were submitted to the open-access repository Zenodo with the Digital Object Identifier (DOI) https://doi.org/10.5281/zenodo.5793891. The cryo-EM density maps and the corresponding model were deposited in the Electron Microscopy Data Bank under accession

numbers EMD-14119 and PDB ID 7QPD. Source data are provided with this paper.

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

## Acknowledgements

This work was supported by the German Research Foundation (TA157/12-1 and CRC 1507 to R.T.) and the European Research Council (ERC Advanced

Grant No. 789121 to R.T.). We thank Dr. Erich Stefan, Inga Nold, Andrea Pott, and all members of the Institute of Biochemistry (Goethe University Frankfurt) for helpful advice and comments. We are grateful to Drs. Achilleas Frangakis and Anja Seybert (Goethe University Frankfurt) for access to the cryo-EM infrastructure via the Frankfurt Center of Electron Microscopy (FCEM). Data processing was carried out in the cryo-EM facility of the Institute of Biochemistry, Goethe University Frankfurt and the Research Center SFB 1507 (Z02 – high-resolution cryo-EM infrastructure) funded by the German Research Foundation (CRC 1507 to R.T.).

## Author contributions

Cell culture and PLC preparations were devised by A.D. and L.H. A.D. prepared the cryo-EM samples and collected the fluorescence polarization data. C.W. carried out the LC-MS analyses, the GluII assays, and peptide synthesis. A.D., L.S., and C.T. analyzed the cryo-EM data and built the final PLC model. A.D., C.W., C.T., S.T., and R.T. prepared the figures and wrote the manuscript. M.H. and N.Z. provided the GluII. Study and experiments were designed by A.D., C.W., and R.T. The study was supervised and conceived by R.T.

## Funding

## Competing interests

The authors declare no competing interests.
