## [Peer Review File · Nature Communications]

Molecular basis of MHC I quality control in the peptide loading complexREVIEWER COMMENTS

Reviewer #1 (Remarks to the Author):

There are some very interesting confirmatory results generated in a new setting. Here and as always this group demonstrates exceptional technical prowess to achieve them. This reviewer would contend however that the key findings are incremental and that they are not set in the context of what is already known about the molecular basis of peptide editing in the PLC.

- The connection between peptide loading & editing in the PLC and glucose trimming is nicely demonstrated, and the authors can reasonably claim that they show the intersection between tapasin-assisted peptide editing and MHC protein quality control. However, the authors imply that peptide editing (ie tapasin assisted peptide exchange) is somehow mechanistically dependent on MHC I HC glycan processing, which it is not. There is much in the literature on this. Glycan processing is almost certainly involved in the recycling of suboptimally loaded MHC I (in the literature) but the current system analyses events in vitro that does not accommodate intracellular MHC I trafficking between compartments. The sentence “our glycan processing assay revealed the MHC I glycan is critical for sensing optimally loaded high affinity peptides in the PLC...” is misleading to the reader who is unfamiliar with the literature.

They describe a scenario that is more in tune with the literature in the figure legend for fig 4, but it seems at odds with the highlighted sentence.

- The “editing loop” of tapasin. A slightly better name than the scoop loop, but again, one that implies a function that is not demonstrated. From other work documented in the literature, this loop is unlikely to mediate editing in isolation – the beta hairpin and multiple interactions shared by MHC-I and tapasin, and other PLC members, are more likely to be synergistically responsible for peptide editing than just the hypothetical loop. It’s nice to see the electron density in the figure, and how their modelled structure fits this or not. The proposed role of Tpn Leu18, forcing MHC Tyr84 to form another interaction (with Tpn Glu72), with Leu18 displacing both the alpha 1 helix and alpha 2-1 helix is compelling, and interesting how this might differ with TAPBPR - something that the authors do not discuss. The potential interaction between MHC-I Y84 and Tpn E72 is likely non-essential, as noted extensively in the literature. The disposition of the editing loop is strikingly reminiscent of the McShan 2020 paper, which the authors forgot to mention.

- the editing assay is not an editing assay. It's a peptide dissociation expt, in which they are simply measuring the dissociation rate of the labelled peptide. The binding of the fluorescent peptide was also quite modest, and the data do not suggest an average of multiple replicates. There is extensive literature on peptide-triggered dissociation of tapasin upon formation of a stable pMHC (including the authors own) which cannot be demonstrated with the nanodisc platform which enforces molecular proximity. As such, the biochemistry is non-physiological. They referred to the importance of CRT remaining within this peptide-loaded PLC complex, saying this showed GluII could trim the glycan while CRT was present. However, the pMHC would normally be expected to dissociate from the core PLC in vivo.

The authors make reference to a single side-chain interaction that they contend could be involved in MHC allele-specific dependency on tapasin binding with no reference to the large body of evidence in the literature that addresses this issue: which is physiologically of great importance in determining whether an individual will mount a protective T cell response to pathogens.

- concerning figure 3a and extended figure 6. The change in mass is identical whether it's a mannose that is removed (e.g. ext 6a Glc1Man8GlcNAc2 = 40178), or a Glucose that is removed (e.g. ext 6b Man9GlcNAc2 = 40178). So, while their interpretation of the change in mass makes sense when specific peptide is present (extended figure 6B), especially considering the specificity GluII, it is disappointing that there is no formal confirmation of glu and not man as the leaving group.

Reviewer #2 (Remarks to the Author):

This is a nice study by Domnick, Winter et al. The noteworthy results are a new higher resolution cryo-EM structure of the PLC relative to their previously reported PLC structure (reference 13; Blee et al). The higher resolution structure resolves more details of the MHC-I glycan interactions with calreticulin as well as of the tapasin-MHC-I interaction. Mass spectrometry is used to elucidate the nature of the MHC-I and tapasin glycans within the PLC and reveal the presence of monoglucosylated MHC-I as predominant MHC-I glycoform. The Glucosidase sensitivity of the MHC-I glycan is further elucidated via mass spectrometry, under different conditions, to reveal allosteric coupling between high affinity peptide binding and glycan processing by Glucosidase II. Overall, the methodology is sound and the studies advance our understanding of the PLC interactions and dynamics.

This conclusion should be further solidified:

"We did not observe any change in size and overall composition of the PLC upon

peptide loading of MHC I and glycan trimming by Glul1 (Fig. 3d, Extended Data Fig. 7), indicating that MHC I glycan processing by Glul1 occurs upon loading of optimal epitopes in the fully assembled PLC containing calreticulin (Fig. 3e)." Fig 3e, should be analyzed by immunoblots similar to Figure 1d, to support this conclusion. In fact, extended Figure 7 indicates a shoulder between 3 and 3.5 ml in the +GlcII+CFSN condition, which could be indicative of protein dissociation from the PLC (are either calreticulin/tapasin or both substoichiometric in this condition-it is hard to deduce this point based on just the coomassie staining gel shown in Fig. 3e).

Did the authors attempt a cryo EM structure following peptide addition to the PLC?

Other points that should be discussed are the identity and HLA genotype of the Burkitt lymphoma cells from which the PLC is purified and why HLA-A*03:01 is the predominant allomorph that was isolated.

What are the expected molecular weights of the other allomorphs with and without PNGase treatments, and is there evidence for their presence? If not, why not?

The authors should clarify differences in interactions mediated by the editing loop of tapasin and the parallel loop of TAPBPR, since there were differences in the loop placements based on the two TAPBPR structures (refs 25 and 26).

Malini Raghavan

Reviewer #3 (Remarks to the Author):

The authors shown the allosteric coupling of glycan processing with peptide proofreading in the PLC using protein chemistry, mass spectrometry, fluorescent polarization, and cryoEM. The manuscript is clear, illustration are great and methods are detailed. The structure is showing the interaction between multiple subunit: tapasin, MHC hc+b2m, ERp57, and calreticulin, and show the mechanism of interaction between them. In addition, the study also reveal the role of glycan and how glycosylation is critical in the quality control of the final peptide-MHC I complex formation.

The observation of the Lys16 of the editing loop flexibility, instead of stabilising the F pocket of HLA-A3 is also clarify the fact that the Lys16 is not restricted to specific acidic F pocket containing HLA molecules, which would have restricted its role and maybe even be an issue for other HLAs.

Line 14.

“MHC I glycan” is a unusual wording I believe and does sounds like MHC I is a glycan. I would suggest to rephrase to clarify that it is the glycan on the MHC I molecule.

Line 31.

I would suggest to replace “immunodominant” by simply immunogenic, as the dominance is irrelevant for a given efficient immune response.

Line 48.

hc is not defined.

Fig 1b.

The hc band seems smaller in intensity than the b2m, is this correct? And why this would occur?

Fig1c.

The composition of the glycans on MHC-I is demonstrated, with the main HLA here been HLA-A3. Do the authors know or tested other HLA? And would this be different for non-classical HLA Ib, or MHC-like molecule such as the CD1 or MR1?

Line 97.

On fig2a we can't see the Cys95-Cys33 disulfide bond between tapasin and ERp57, maybe add a “*” on the figure to indicate its location and have a close up fig in supp.

Line 126.

It is interesting that some contacts are made with HLA residue that are not conserved in HLA molecules (like Arg187), would this make some HLA molecules better equipped/favoured to be loaded with peptides through the PLC?

Fig2f.

For the empty and peptide loaded MHC, is the same MHC used? If no this is an important consideration. If yes did you use multiple structure of the peptide-MHC as some allomorph have a lot of flexibility in their cleft depending on the bound peptide.

Line 152.

It is unclear if the suboptimal R9L peptide bind at all to HLA-A3, as the primary anchor residues are different from the CSN peptide. If the R9L peptide does not bind it might not be a good control here. Would R9L be equivalent to just buffer? And on line 159 it's written that the peptide is a no-binder.

Reviewer #4 (Remarks to the Author):

In this manuscript, Domnick, Winter et al. purified the peptide loading complex (PLC), consisting of the heterodimeric peptide TAP transporter, together with client MHC I and its associated chaperones tapasin, calreticulin and ERp57, and reconstituted it into nanodiscs. LC-MS analysis established that the majority of the MHC I molecules carried a Glc1Man9GlcNAc2 glycan (while tapasin glycosylation was heterogeneous). The authors then determined the structure of the nanodisc-reconstituted PLC by cryoEM to a maximum resolution of 3.7 Å, which confirmed major structural characteristics previously observed in a structure for the PLC in detergent. In particular, the map confirmed that the PLC is organized in two editing modules with one editing module being fully assembled and the other one lacking some of the components. The map and the built model resolved several new features. The MHC I glycan was resolved and is seen to interact with the calreticulin lectin domain. A tapasin loop, referred to as editing loop, would clash with peptide-bound MHC I and is thus thought to contribute to peptide exchange catalysis. The MHC I adopts an open conformation and the authors speculate that closing of the MHC I peptide-binding groove upon peptide editing might result in the release of the MHC I glycan from the lectin domain of calreticulin. After establishing the peptide proofreading activity of nanodisc-reconstituted PLC by fluorescence polarization, the authors use LC-MS analysis to show that the terminal glucose of MHC I is only cleaved by Glu1 if it is occupied by a high-affinity peptide, thus licensing the pMHC I to move to the cell surface. The authors also provide evidence that glycan trimming by Glu1 does not require calreticulin to dissociate from the PLC. Based on their findings, the authors propose a model for coupled peptide loading and quality control of MHC I.

This is an elegant study that provides clear evidence as well as some structural foundation for the coupling of MHC I peptide editing with the licensing of MHC I for trafficking to the cell surface. The mechanistic basis for this coupling in the PLC may well be representative for other quality control processes in the ER, thus further increasing the importance of this study. Overall, I fully support publication of this beautiful work. I only have a few points that the authors might be able to comment/speculate on, noting that I would not expect the authors to address these points experimentally.

The evidence for the necessity for MHC I to be loaded with a high-affinity peptide for glycan trimming to occur is very strong. The authors then hypothesize “that the closing of the peptide-binding groove after successful peptide editing might result in a release of the glycan from the lectin domain of calreticulin”. However, the closure of the MHC I peptide-binding groove seems a bit subtle and one could imagine that if one would model the PLC with peptide-bound MHC I in the closed conformation, it would still be possible for the glycan to associate with calreticulin. Can the authors expand on their speculation on how peptide binding might result in the glycan to become accessible to GluI?

Similarly, it seems to be an established fact (this reviewer is not an expert in this field) that trimming of the final glucose licenses MHC I to be trafficked to the Golgi. In the model in Figure 4, the authors propose that glycan trimming results in the liberation of the pMHC I from the PLC. Can the authors speculate on how the removal of a single terminal glucose from a glycan (that has already been released from calreticulin) would allow pMHC I to dissociate from the PLC? Along the same line, can the authors summarize for non-experts what is known about how the PLC retains its bound MHC I in the ER whereas pMHC I can be trafficked to the Golgi? As stated above, no experiments are needed, just some additional discussion and potentially speculation.

The authors used CryoDRGN to analyze their dataset and found maps that they state represent assembly intermediates. These maps that are presented in Extended Data Figure 4 are not really analyzed in any way. If the authors flexibly fit the components of the PLC into these maps, what PLC components are these assembly intermediates missing? How do these assembly intermediates relate to the model shown in Figure 4?

Some minor points:

Abstract: The abbreviation “PLC” for “peptide loading complex” needs to be defined.

Page 4, line97: The reader is referred to Figure 2a for the disulfide bond between tapasin Cys95 and ERp57 Cys33, but this bond is not obvious in the figure.

The data processing workflow depicted in Extended Data Figure 3 is a bit too short. All the 2D and 3D classification steps that were used to obtain the final maps for the full PLC and the single editing module from the respective initial models should be presented in figure and text.

Similarly, since CryoDRGN is still very new, the text, which is very short, should provide more details for the CryoDRGN analysis. Extended Data Figure 4 should present all the steps in figure and text, rather than to just denote “cryoDRGN analysis; cryoSPARC 3D classification and homogeneous refinement”. The figure caption also needs to explain the panels shown below the 3D maps.

Extended Data Figure 4 appears to show the local resolution map and 2D averages for the map of the single editing module, whereas the FSC is shown for the full PLC and it is unclear for which map the angular assignment of particles is shown. The local resolution map, 2D averages, FSC curves and angular assignment should be shown for both maps. Comparisons of the model with density should also be shown for both maps, especially for de novo modeled regions.

POINT-TO-POINT REPLY

Reviewer #1:

There are some very interesting confirmatory results generated in a new setting. Here and as always, this group demonstrates exceptional technical process to achieve them. This reviewer would contend however that the key findings are incremental and that they are not set in the context of what is already known about the molecular basis of peptide editing in the PLC.

Reply: We wish to thank the reviewer for this positive and encouraging feedback. We respectfully disagree with the statement that the key findings are incremental. Our work addresses fundamental aspects of how pMHC I maturation is linked to glycan processing and the calnexin/calreticulin cycle of ER quality control (ERQC). Based on the 3.7-Å structure of the PLC editing module in a near-native lipid environment, we are able to describe the unique glycan/protein-protein interaction network that stabilizes peptide-receptive MHC I clients. Furthermore, using a reconstituted peptide editing and deglycosylation system, we demonstrate that the key ERQC enzyme glucosidase II can act on the fully assembled PLC, with the glycan trimming being allosterically coupled to peptide proofreading. At this point, we would also like to refer to the positive assessments of the other reviewers (e.g., reviewer #2: “...the studies advance our understanding of the PLC interactions and dynamics.”; reviewer #4: “...the built model resolved several new features...”, “...coupling in the PLC may well be representative for other quality control processes in the ER, thus further increasing the importance of this study...”).

- 1) *The connection between peptide loading and editing in the PLC and glucose trimming is nicely demonstrated, and the authors can reasonably claim that they show the intersection between tapasin-assisted peptide editing and MHC protein quality control. However, the authors imply that peptide editing (ie tapasin assisted peptide exchange) is somehow mechanistically dependent on MHC I HC glycan processing, which it is not. There is much in the literature on this. Glycan processing is almost certainly involved in the recycling of suboptimally loaded MHC I (in the literature) but the current system analyses events in vitro that does not accommodate intracellular MHC I trafficking between compartments. The sentence “our glycan processing assay revealed the MHC I glycan is critical for sensing optimally loaded high affinity peptides in the PLC...” is misleading to the reader who is unfamiliar with the literature. They describe a scenario that is more in tune with the literature in the figure legend for fig 5, but it seems at odds with the highlighted sentence.*

Reply and Action Taken: Thank you for this helpful comment. We rephrased the sentence accordingly: “Our glycan processing assay showed that, as far as the nanodisc-reconstituted PLC is concerned, the status of the MHC I glycan is coupled with the peptide loading status”.

- 2) *The “editing loop” of tapasin. A slightly better name than the scoop loop, but again, one that implies a function that is not demonstrated. From other work documented in the literature, this loop is unlikely to mediate editing in isolation – the beta hairpin and multiple interactions shared by MHC-I and tapasin, and other PLC members, are more likely to be synergistically responsible for peptide editing than just the hypothetical loop. It’s nice to see the electron density in the figure, and how their modelled structure fits this or not. The proposed role of Tpn Leu18, forcing MHC Tyr84 to form another interaction (with Tpn Glu72), with Leu18 displacing both the alpha 1 helix and alpha 2-1 helix is compelling, and interesting how this might differ with TAPBPR - something that the authors do not discuss. The potential interaction between MHC-I Y84 and Tpn E72 is likely non-essential, as noted extensively in the literature. The disposition of the editing loop is strikingly reminiscent of the McShan 2020 paper, which the authors forgot to mention.*

Reply: Thank you for your comment on the editing loop. We agree that the editing loop does not mediate peptide editing in isolation from either tapasin or TAPBPR. As shown in the Fig. 2d and 2e, the β-hairpin below the binding groove, as well as the extensive interface between tapasin and MHC I contribute to the editing process. However, the editing loop is in direct contact with the peptide binding site in the F-pocket, and most likely with putative cargo peptides, which is not the

case for any of the other editing features of MHC I chaperones. Therefore, we believe that labelling the editing loop as such is an appropriate description.

We wish to thank the reviewer for pointing out the missing (graphical) comparison with the TAPBPR binding loop. In the revised manuscript, we now compared the tapasin editing loop with the editing loop of the TAPBPR-H2-D^b crystal structure (new Suppl. Fig. 6b). The loop of TAPBPR is considerably longer than the tapasin loop (16 vs. 10 residues). Both loops interact with MHC I in the F-pocket region. While tapasin uses Leu18 to disturb the interaction between MHC I Tyr84 and the peptide C terminus, this task is performed by residues 34-36 in TAPBPR. Furthermore, the extended architecture in TAPBPR causes the loop to dip into the peptide binding groove to facilitate chaperoning and peptide exchange. The posture of the loop recently described by McShan *et al.* 2021, resembles the characteristics found in the cryo-EM density. The density in the loop region is not as well defined as the surrounding regions, indicating a relative flexibility of the loop. The reference McShan *et al.* has been added as reference.

- 3) *The editing assay is not an editing assay. It's a peptide dissociation expt, in which they are simply measuring the dissociation rate of the labelled peptide. The binding of the fluorescent peptide was also quite modest, and the data do not suggest an average of multiple replicates. There is extensive literature on peptide-triggered dissociation of tapasin upon formation of a stable pMHC (including the authors own) which cannot be demonstrated with the nanodisc platform which enforces molecular proximity. As such, the biochemistry is non-physiological. They referred to the importance of CRT remaining within this peptide-loaded PLC complex, saying this showed Glu11 could trim the glycan while CRT was present. However, the pMHC would normally be expected to dissociate from the core PLC in vivo.*

The authors make reference to a single side-chain interaction that they contend could be involved in MHC allele-specific dependency on tapasin binding with no reference to the large body of evidence in the literature that addresses this issue: which is physiologically of great importance in determining whether an individual will mount a protective T cell response to pathogens.

Reply: Thank you for your comment. We agree that the nanodisc enforces molecular proximity. We concluded that this proximity allowed tapasin to keep close contact to MHC I to facilitate exchange of the high-affinity binder C^FSN to the thermodynamically favored competitor CSN. We observed that maximum peptide binding to PLC-associated MHC I was reached within a few minutes (Fig. 3a), which is faster than peptide binding to HLA-A*03:01 alone or catalyzed by tapasin-ERp57 or TAPBPR using similar conditions (Lan *et al.* 2021). This facilitated exchange demonstrates the peptide-editing function of tapasin in a fully assembled peptide loading complex. Moreover, we show that Glu11 can trim the terminal glucose from pMHC I. In this case, the nanodisc enforces the proximity, and we observe both peptide loading and glycan trimming. Glycan trimming is possible once the peptide is loaded. Presumably, simultaneously the interactions between tapasin and pMHC I are weakened, eventually causing the release of pMHC I from the PLC. To study this process in detail, an unlimited membrane system, which allows pMHC I release, is necessary. However, reconstitution of an intrinsically transient supramolecular assembly like the PLC (650 kDa) into liposomes will be a major challenge. We rephrased the discussion accordingly.

As far as allomorph specificity is concerned, sequence alignments of several classical MHC I molecules (tapasin-dependent and -independent) show that the interface between tapasin and MHC I is conserved on MHC I. However, Asn127 is not conserved and might be important for client specificity. Yet, we are very cautious in phrasing the corresponding sentence (“...might contribute to substrate specificity...”), because we agree with the reviewer that allomorph specificity of the peptide editors is an intricate problem that cannot be reduced to single residues but most likely involves the intrinsic dynamics of different allomorphs.

AA Pos.	110	120	130	140	150
A*03:01:01:01	CDVGS DGRFL	RGYRQ DAYDG	KDYIALNEDL	RSWTAADMAA	QITKRKWEAA
A*01:01:01:01	----P-----	-----	-----	-----	-----V
A*01:02:01:01	----P-----	-----	-----	-----	-----V
A*02:01:01:01	-----W---	---H-Y---	-----K---	-----	-T--H----
A*02:02:01:01	-----W---	---H-Y---	-----K---	-----	-T--H----
A*02:05:01:01	-----W---	---H-Y---	-----K---	-----	-T--H----
A*68:01:01:01	-----	-----	-----K---	-----	-T--H----
A*80:01:01:01	-----	-----	-----K---	-----	-T--H----

- 4) *Concerning Figure 3a and Suppl. Figure 7. The change in mass is identical whether it's a mannose that is removed (e.g. Suppl. Fig. 7a Glc1Man8GlcNAc2 = 40178), or a Glucose that is removed (e.g. Suppl. Fig. 7b Man9GlcNAc2 = 40178). So, while their interpretation of the change in mass makes sense when specific peptide is present (Suppl. Fig. 7B), especially considering the specificity Glull, it is disappointing that there is no formal confirmation of Glu and not man as the leaving group.*

Reply: Since mannose and glucose are epimers, the determination of the leaving group by intact protein mass spectrometry is not possible. However, since Glull exclusively cleaves terminal glucose residues at the A branch, with flexibility in the number of mannoses at the B and C branches (Totani *et al.* 2006 JBC), our conclusion is the only plausible one. The structure of the glycans we described is supported by Radcliffe *et al.* 2002, where arm-specific isomers were distinguished.

Reviewer #2:

This is a nice study by Domnick, Winter et al. The noteworthy results are a new higher resolution cryo-EM structure of the PLC relative to their previously reported PLC structure (reference 13; Blees et al). The higher resolution structure resolves more details of the MHC-I glycan interactions with calreticulin as well as of the tapasin-MHC-I interaction. Mass spectrometry is used to elucidate the nature of the MHC-I and tapasin glycans within the PLC and reveal the presence of monoglucosylated MHC-I as predominant MHC-I glycoform. The Glucosidase sensitivity of the MHC-I glycan is further elucidated via mass spectrometry, under different conditions, to reveal allosteric coupling between high affinity peptide binding and glycan processing by Glucosidase II. Overall, the methodology is sound, and the studies advance our understanding of the PLC interactions and dynamics.

Reply: We wish to thank the reviewer for these very positive comments highlighting the major advancements in the mechanistic understanding of interactions and dynamics within the PLC and their integration with ER quality control.

- 1) *This conclusion should be further solidified: "We did not observe any change in size and overall composition of the PLC upon peptide loading of MHC I and glycan trimming by Glull (Fig. 3d, Extended Data Fig. 7), indicating that MHC I glycan processing by Glull occurs upon loading of optimal epitopes in the fully assembled PLC containing calreticulin (Fig. 3e)." Fig. 3e, should be analyzed by immunoblots similar to Figure 1d, to support this conclusion. In fact, Suppl. Figure 7 indicates a shoulder between 3 and 3.5 ml in the +GlcII+CFSN condition, which could be indicative of protein dissociation from the PLC (are either calreticulin/tapasin or both sub-stoichiometric in this condition-it is hard to deduce this point based on just the Coomassie staining gel shown in Fig. 3e).*

Reply: Thank you for the helpful suggestion. We have analyzed the PLC components calreticulin, HLA-A, tapasin, β_2m , and TAP1 to monitor the PLC composition before and after peptide loading and glycan trimming by SDS-PAGE and immunoblotting (new Fig. 4). Relative to the core PLC component TAP1, we did not observe a change in the amount of calreticulin. Considering the almost complete deglycosylation of HLA-A*03:01 (Fig. 3b) by Glull after peptide loading, we are confident that the presence of calreticulin in the PLC does not interfere with glycan trimming. In addition, we analyzed the shoulder region between 3.0 mL and 3.5 mL in all PLC-containing conditions (Fig. R1). Minute amounts of calreticulin were detected in the shoulder fraction

(position *), including the untreated PLC samples. Almost no calreticulin appears to be eluted in the presence of GluII and C^FSN peptide. Overall, the amount of free calreticulin is very low, compared to the amount present in the assembled PLC. None of the membrane protein components of the PLC (HLA-A, tapasin, and TAP1) were observed in free form in the shoulder region.

Fig. R1: Immunoblot analysis of PLC after peptide loading and MHC I deglycosylation. **a** The composition of Nd-reconstituted PLC remains unchanged after peptide binding and GluII trimming as demonstrated by SEC (Fig. 4a). PLC and GluII peak fractions were analyzed by SDS-PAGE (Fig. 4b) and immunoblotting (Fig. 4c). **b** Immunoblotting of the shoulder region (*) between 3.0 and 3.5 mL in PLC containing samples after treatment with peptide and deglycosylation by GluII.

2) *Did the authors attempt a cryo EM structure following peptide addition to the PLC?*

Reply: Structural analysis of PLC::MSP2N2 in the presence of allomorph-specific peptide has been attempted but screening results were so far discouraging. The presence of the peptide seems to render the PLC more dynamic, and structure analysis in the presence of high-affinity binders has not been successful so far.

3) *Other points that should be discussed are the identity and HLA genotype of the Burkitt lymphoma cells from which the PLC is purified and why HLA-A*03:01 is the predominant allomorph that was isolated. What are the expected molecular weights of the other allomorphs with and without PNGase treatments, and is there evidence for their presence? If not, why not?*

Reply: The Raji cells contain the three classical HLA allomorphs (A*03:01: MW 40,339.9 Da, MW_{PNGase}: 38,312.1 Da; B*15:10: m/z: 39,921.3 Da, MW_{PNGase}: 37,893.5 Da; C*04:01 m: 40,408.0 Da, MW_{PNGase}: 38,380.2 Da) as well as the non-classical allomorphs E/F/G (Blees *et al.* 2017 *Nature*; Sethumadhaven *et al.* 2022 *Sci Rep*). The HLA-B and HLA-C allomorphs have been observed in significantly lower abundance than HLA-A via LC-MS/MS (Sethumadhavan *et al.* 2022 *Sci Rep*). Non-classical MHC I molecules are recruited to the PLC in very low abundance so that they are not

detected by intact protein MS. In this study, HLA-B and HLA-C were identified by intact protein LC-MS, but in amounts close to background noise, preventing further analysis.

- 4) *The authors should clarify differences in interactions mediated by the editing loop of tapasin and the parallel loop of TAPBPR, since there were differences in the loop placements based on the two TAPBPR structures (refs 25 and 26).*

Reply: We wish to thank the reviewer for pointing this out (see also Reviewer #1 Point 2). We have added a new Supplementary Fig. 6b, which compares the tapasin editing loop with the editing loop of the TAPBPR-H2-D^b crystal structure. The loop from TAPBPR is considerably longer than the tapasin loop (16 vs. 10 residues). Both loops interact with MHC I in the F-pocket region, especially near the α 1 helix. While tapasin uses Leu18 to disturb the interaction between MHC I Tyr84 and the peptide C-terminus, this task is performed by residues 34-36 in TAPBPR. Furthermore, the extended architecture in TAPBPR causes the loop to dive into the peptide binding groove to facilitate chaperoning and peptide exchange.

Reviewer #3:

The authors shown the allosteric coupling of glycan processing with peptide proofreading in the PLC using protein chemistry, mass spectrometry, fluorescent polarization, and cryo-EM. The manuscript is clear, illustrations are great, and methods are detailed. The structure is showing the interaction between multiple subunit: tapasin, MHC hc+b2m, ERp57, and calreticulin, and show the mechanism of interaction between them. In addition, the study also reveals the role of glycan and how glycosylation is critical in the quality control of the final peptide-MHC I complex formation. The observation of the Lys16 of the editing loop flexibility, instead of stabilizing the F pocket of HLA-A3 is also clarify the fact that the Lys16 is not restricted to specific acidic F pocket containing HLA molecules, which would have restricted its role and maybe even be an issue for other HLAs.

Reply: We thank the reviewer for the encouraging comments highlighting the advancements in mechanistic understanding of the peptide loading and quality control process in the PLC.

- 1) *Line 14. "MHC I glycan" is a unusual wording I believe and does sounds like MHC I is a glycan. I would suggest to rephrase to clarify that it is the glycan on the MHC I molecule.*

Reply and Action Taken: Thank you for your comment. We have defined MHC I glycan in the text: Besides the structurally defined N-linked, hereafter referred to simply as "MHC I glycan" [...].

- 2) *Line 31. I would suggest to replace "immunodominant" by simply immunogenic, as the dominance is irrelevant for a given efficient immune response.*

Reply: Thank you for this comment. Changed accordingly.

- 3) *Line 48. Hc is not defined.*

Reply: Changed accordingly.

- 4) *Fig 1b. The hc band seems smaller in intensity than the b2m, is this correct? And why this would occur?*

Reply: The intensity of signal in immunoblots depends not only on the amount of the investigated protein but also on the quality and titer of the antibodies used. Different antibodies had to be used for MHC I (HC10, Acris Antibodies, catalogue number AM33035PU-N) and β_2m (Novo Antibodies, catalogue number HPA006361). The Coomassie-stained SDS-PAGE gels of Fig. 1b and Fig. 3e show expected ratios of MHC I to β_2m . Here, a lower intensity is expected due to the lower molecular mass of β_2m and the resulting lower intensity in Coomassie staining.

- 5) *Fig1c. The composition of the glycans on MHC-I is demonstrated, with the main HLA here been HLA-A3. Do the authors know or tested other HLA? And would this be different for non-classical HLA Ib, or MHC-like molecule such as the CD1 or MR1?*

Reply: The Raji cells contain the three classical HLA allomorphs (A*03:01: MW 40,339.9 Da, MW_{PNGase}: 38,312.1 Da; B*15:10: m/z: 39,921.3 Da, MW_{PNGase}: 37,893.5 Da; C*04:01 m: 40,408.0 Da, MW_{PNGase}: 38,380.2 Da) as well as the non-classical allomorphs E/F/G (Blees *et al.* 2017 *Nature*; Sethumadhaven *et al.* 2022 *Sci Rep*). The HLA-B and HLA-C allomorphs have been observed in significantly lower abundance than HLA-A via LC-MS/MS (Sethumadhavan *et al.* 2022 *Sci Rep*). Non-classical MHC I molecules are recruited to the PLC in very low abundance so that they are not detected by intact protein MS. In this study, HLA-B and HLA-C were identified by intact protein LC-MS, but in amounts close to background noise, preventing further analysis.

- 6) *Line 97. On Fig. 2a we can't see the Cys95-Cys33 disulfide bond between tapasin and ERp57, maybe add a "*" on the figure to indicate its location and have a close-up fig in supp.*

Reply: Thank you for pointing this out. A zoom-in has now been added (new Supplementary Fig. 6a).

- 7) *Line 126. It is interesting that some contacts are made with HLA residue that are not conserved in HLA molecules (like Arg187), would this make some HLA molecules better equipped/favored to be loaded with peptides through the PLC?*

Reply: Thank you for raising this important question (see also point 3 of reviewer #1). As far as allomorph specificity is concerned, sequence alignments of several classical MHC I molecules (tapasin-dependent and -independent) show that the interface between tapasin and MHC I is conserved on MHC I. However, Asn127 is not conserved and might be important for client specificity. Yet, we are very cautious in phrasing the corresponding sentence ("...might contribute to substrate specificity..."), because we agree with the reviewer that allomorph specificity of the peptide editors is an intricate problem that cannot be reduced to single residues but most likely involves the intrinsic flexibility of different allomorphs.

AA Pos.	110	120	130	140	150		
A*03:01:01:01	CDVGS	DGRFL	RGYRQ	DAYDG	KDYIALNEDL	RSWTAADMAA	QITKRKWEAA
A*01:01:01:01	----	P-----	-----	-----	-----	-----	-----V
A*01:02:01:01	----	P-----	-----	-----	-----	-----	-----V
A*02:01:01:01	-----	W----	---H-Y---	-----	K----	-----	-T--H-----
A*02:02:01:01	-----	W----	---H-Y---	-----	K----	-----	-T--H-----
A*02:05:01:01	-----	W----	---H-Y---	-----	K----	-----	-T--H-----
A*68:01:01:01	-----	-----	-----	-----	K----	-----	-T--H-----
A*80:01:01:01	-----	-----	-----	-----	-----	-----	-----

Fig. 2f. For the empty and peptide loaded MHC, is the same MHC used? If no this is an important consideration. If yes, did you use multiple structure of the peptide-MHC as some allomorphs have a lot of flexibility in their cleft depending on the bound peptide.

Reply: Yes, we used the same MHC I allomorph (HLA-A*03:01) for the comparison in Fig. 2f (PDB ID of peptide-bound form: 3RL1, Zhang *et al.* 2011). However, we also performed alignments with several additional peptide-bound HLA-A*03:01 structures from the PDB. These structures contain different peptides but exhibit only minor deviations from the peptide-bound structure we show in Fig. 2f, as illustrated in Fig. R2 below.

Fig. R2: Superposition of peptide-loaded MHC I with empty MHC I in the PLC. Overlay of pMHC I (HLA-A*03:01) structures with the empty HLA-A*03:01 in the PLC. Teal - empty - this publication, Khaki - loaded – PDB ID: 3RL1, Red - loaded - PDB ID: 2XPG, Orange - loaded - PDB ID: 7L1C, Cyan - loaded - PDB ID: 7L1B, Purple - loaded - PDB ID: 7MLE, Yellow - loaded - PDB ID: 609C, Blue - loaded - PDB ID: 609B.

- 8) *Line 152. It is unclear if the suboptimal R9L peptide bind at all to HLA-A3, as the primary anchor residues are different from the CSN peptide. If the R9L peptide does not bind it might not be a good control here. Would R9L be equivalent to just buffer? And on line 159 it's written that the peptide is a no-binder.*

Reply: Thank you for pointing this out. R9L has a 2000-fold lower affinity for HLA-A*03:01 (20 μ M for R9L vs. 10 nM for CSN, according to NetMHC 4.1). We confirmed by fluorescence polarization that the labeled R9L peptide (C4F) does not bind the HLA allomorphs within the PLC. In our view, the suboptimal (non-binding) peptide is a better control than just buffer, because it shows that Glu11 trimming is not induced by any peptide, but only by high-affinity binders, demonstrating the coupling of Glu11 trimming with peptide editing.

Reviewer #4:

In this manuscript, Domnick, Winter et al. purified the peptide loading complex (PLC), consisting of the heterodimeric peptide TAP transporter, together with client MHC I and its associated chaperones tapasin, calreticulin and ERp57, and reconstituted it into nanodiscs. LC-MS analysis established that the majority of the MHC I molecules carried a Glc1Man9GlcNAc2 glycan (while tapasin glycosylation was heterogeneous). The authors then determined the structure of the nanodisc-reconstituted PLC by cryo-EM to a maximum resolution of 3.7 Å, which confirmed major structural characteristics previously observed in a structure for the PLC in detergent. In particular, the map confirmed that the PLC is organized in two editing modules with one editing module being fully assembled and the other one lacking some of the components. The map and the built model resolved several new features. The MHC I glycan was resolved and is seen to interact with the calreticulin lectin domain. A tapasin loop, referred to as editing loop, would clash with peptide-bound MHC I and is thus thought to contribute to peptide exchange catalysis. The MHC I adopts an open conformation, and the authors speculate that closing of the MHC I peptide-binding groove upon peptide editing might result in the release of the MHC I glycan from the lectin domain of calreticulin. After establishing the peptide proofreading activity of nanodisc-reconstituted PLC by fluorescence polarization, the authors use LC-MS analysis to show that the terminal glucose of MHC I is only cleaved by Glu11 if it is occupied by a high-affinity peptide, thus licensing the pMHC I to move to the cell surface. The authors also

provide evidence that glycan trimming by Glul1 does not require calreticulin to dissociate from the PLC. Based on their findings, the authors propose a model for coupled peptide loading and quality control of MHC I.

This is an elegant study that provides clear evidence as well as some structural foundation for the coupling of MHC I peptide editing with the licensing of MHC I for trafficking to the cell surface. The mechanistic basis for this coupling in the PLC may well be representative for other quality control processes in the ER, thus further increasing the importance of this study. Overall, I fully support publication of this beautiful work. I only have a few points that the authors might be able to comment/speculate on, noting that I would not expect the authors to address these points experimentally.

Reply: We wish to thank the review for her/his encouraging and very positive comments.

- 1) The evidence for the necessity for MHC I to be loaded with a high-affinity peptide for glycan trimming to occur is very strong. The authors then hypothesize “that the closing of the peptide-binding groove after successful peptide editing might result in a release of the glycan from the lectin domain of calreticulin”. However, the closure of the MHC I peptide-binding groove seems a bit subtle, and one could imagine that if one would model the PLC with peptide-bound MHC I in the closed conformation, it would still be possible for the glycan to associate with calreticulin. Can the authors expand on their speculation on how peptide binding might result in the glycan to become accessible to Glul1?*

Reply: Indeed, the closing of the MHC I peptide-binding groove seems to be a subtle movement. However, the glycan displays an extended conformation, stretched between the MHC I Asn86 and the calreticulin binding groove, as indicated by the relatively high resolution (in contrast to the B/C branch of the glycan, which are not resolved). The subtle movement might be sufficient to disturb the binding of the glycan to calreticulin.

- 2) Similarly, it seems to be an established fact (this reviewer is not an expert in this field) that trimming of the final glucose licenses MHC I to be trafficked to the Golgi. In the model in Figure 4, the authors propose that glycan trimming results in the liberation of the pMHC I from the PLC. Can the authors speculate on how the removal of a single terminal glucose from a glycan (that has already been released from calreticulin) would allow pMHC I to dissociate from the PLC? Along the same line, can the authors summarize for non-experts what is known about how the PLC retains its bound MHC I in the ER whereas pMHC I can be trafficked to the Golgi? As stated above, no experiments are needed, just some additional discussion and potentially speculation.*

Reply: Thank you for your question. In general, the MHC I is held in place within the PLC through its interaction with calreticulin (via its N-linked glycan on Asn86) and its extended interface with tapasin. We propose that glycan trimming on pMHC I is associated with high-affinity peptide binding to MHC I. It is known that high-affinity peptide binding drastically weakens the interaction between the editor tapasin and MHC I, due to peptide-induced conformational changes in the MHC I. This, in combination with the dissolved interaction between the glycan and calreticulin, allows the pMHC I to leave the PLC and travel to the cell surface. Removal of the glucose might be the last necessary step for full release of pMHC I from the PLC.

- 3) The authors used CryoDRGN to analyze their dataset and found maps that they state represent assembly intermediates. These maps that are presented in Extended Data Figure 4 are not really analyzed in any way. If the authors flexibly fit the components of the PLC into these maps, what PLC components are these assembly intermediates missing? How do these assembly intermediates relate to the model shown in Figure 4?*

Reply: We have expanded Supplementary Fig. 4 to illustrate the relationship between the CryoDRGN-derived maps and the different assembly intermediates. The full PLC (I, grey map) is the proposed state of the PLC before/during peptide loading (Fig. 4, 2nd step). The pink map (V) showing a single editing module could be in a similar state, where the density for the nanodisc-encapsulated membrane region is additionally displaced. This map reflects the behavior of a PLC

with a single editing module bending towards the membrane, as observed in molecular dynamics simulations (Fisette *et al.* 2020 PNAS). The yellow, cyan, and purple maps (II, III, IV) appear to represent intermediates that cannot be readily assigned to certain stages of the current model of peptide loading and editing as proposed in Fig. 5. Some particles might get damaged during the cryo-EM freezing process. We have added the composition of the second editing module as a schematic illustration below the final maps in Supplementary Fig. 4.

Some minor points:

- 1) *Abstract: The abbreviation “PLC” for “peptide loading complex” needs to be defined.*

Reply: Thank you. Changed accordingly.

- 2) *Page 4, line97: The reader is referred to Fig. 2a for the disulfide bond between tapasin Cys95 and ERp57 Cys33, but this bond is not obvious in the figure.*

Reply: We wish to thank the reviewer for this helpful suggestion. We have added Supplementary Fig. 6a to illustrate the disulfide bridge between tapasin and ERp57.

- 3) *The data processing workflow depicted in Suppl. Fig. 3 is a bit too short. All the 2D and 3D classification steps that were used to obtain the final maps for the full PLC and the single editing module from the respective initial models should be presented in figure and text.*

Reply: Supplementary Fig. 3 and text have been expanded to illustrate the processing workflow used to obtain the final maps for the full PLC and single editing module, including 2D and 3D classifications, ab-initio models, and intermediate 3D maps.

- 4) *Similarly, since CryoDRGN is still very new, the text, which is very short, should provide more details for the CryoDRGN analysis. Extended Data Figure 4 should present all the steps in figure and text, rather than to just denote “cryoDRGN analysis; cryoSPARC 3D classification and homogeneous refinement”. The figure caption also needs to explain the panels shown below the 3D maps.*

Reply: Supplementary Fig. 4 and the figure caption has been thoroughly revised to explain the cryoDRGN analysis of the PLC, including the PCA projection of latent space encodings by cryoDRGN, ensuring that the figure caption now also explains all panels.

- 5) *Supplementary Figure 4 appears to show the local resolution map and 2D averages for the map of the single editing module, whereas the FSC is shown for the full PLC and it is unclear for which map the angular assignment of particles is shown. The local resolution map, 2D averages, FSC curves and angular assignment should be shown for both maps. Comparisons of the model with density should also be shown for both maps, especially for de novo modeled regions.*

Reply: We thank the reviewer for pointing out this inconsistency. Supplementary Fig. 5 now shows the local resolution map, 2D averages, FSC, and angular assignments of both the full PLC and the single editing module. The figure legend has been modified accordingly. A comparison of the maps and model for the de novo modeled MHC I glycan is included in Fig. 2b. The editing loop and the corresponding map region are depicted in Fig. 2c. Furthermore, Figs. R3 and R4 (see below) show comparisons of the model with both maps, respectively.

Fig. R3: Comparison of the final model of a single editing module with the cryo-EM density of a single editing module. The PLC subunits are color coded: calreticulin (yellow), ERp57 (red), tapasin (orange), MHC I hc (teal), β_2m (green).

Fig. R4: Fitting of the editing module structure into the cryo-EM density of the assembled PLC. The structure of the single editing module was rigid body-docked into the density map of the PLC. Two editing modules could be placed, with one encompassing all subunits, whereas the density of the second editing module only allowed placement of tapasin (blue), MHC I (purple) and β_2m (cyan).

REVIEWERS' COMMENTS

Reviewer #2 (Remarks to the Author):

The majority of my prior concerns have been addressed.

Reviewer #3 (Remarks to the Author):

Thanks to the authors for their careful reply to the questions and comments, and for adding some new figures to further describe the interaction taking place in the complex protein structure study here. I have no further comments.

Reviewer #4 (Remarks to the Author):

The authors have addressed all my concerns and I support publication of this paper.